

# Potential and functional prediction of six circular RNAs as diagnostic markers for colorectal cancer

Li yuan Liu[1,2,*], Dan Jiang[1,2,*], Yuliang Qu[1], Hongxia Wang[1], Yanting Zhang[1], Shaoqi Yang[1], Xiaoliang Xie[1], Shan Wu[1], Haijin Zhou[2] and Guangxian Xu[1,2]

[1] School of Clinical Medicine, The Third Affiliated Hospital of Ningxia Medical University, Ningxia Medical University, Yinchuan, Ningxia, China
[2] Guangdong Provincial Key Laboratory of Medical Molecular Diagnostics, School of Medical Technology, Institute of Clinical Laboratory, Guangdong Medical University, Dongguan, Guangdong, China
[*] These authors contributed equally to this work.

## ABSTRACT

**Background.** Circular RNAs (circRNAs) have been discovered in colorectal cancer (CRC), but there are few reports on the expression distribution and functional mining analysis of circRNAs.

**Methods.** Differentially expressed circRNAs in CRC tissues and adjacent normal tissues were screened and identified by microarray and qRT-PCR. ROC curves of the six circRNAs were analyzed. A series of bioinformatics analyses on differentially expressed circRNAs were performed.

**Results.** A total of 207 up-regulated and 357 down-regulated circRNAs in CRC were screened, and three top up-regulated and down-regulated circRNAs were chosen to be verified in 33 pairs of CRCs by qRT-PCR. 6 circRNAs showed high diagnostic values (AUC = 0.6860, AUC = 0.8127, AUC = 0.7502, AUC = 0.9945, AUC = 0.9642, AUC = 0.9486 for hsa_circRNA_100833, hsa_circRNA_103828, hsa_circRNA_103831 and hsa_circRNA_103752, hsa_circRNA_071106, hsa_circRNA_102293). A circRNA-miRNA-mRNA regulatory network (cirReNET) including six candidate circRNAs, 19 miRNAs and 210 mRNA was constructed, and the functions of the cirReNET were predicted and displayed via Gene Ontology (GO) and Kyoto Encyclopedia of Genes and Genomes (KEGG) analyses on these mRNAs and protein-protein interaction (PPI) network of the hub genes acquired by string and CytoHubba.

**Conclusion.** A cirReNET containing potential diagnostic and predictive indicators of CRCs and several critical circRNA-miRNA-mRNA regulatory axes (cirReAXEs) in CRC were mined, and may provide a novel route to study the mechanism and clinical targets of CRC.

# INTRODUCTION

Colorectal cancer (CRC) is the fourth most deadly cancer worldwide, with almost 900,000 deaths annually (*Li et al., 2020a*). With nearly 2.1 million new cases each year, the incidence of CRC is projected to increase to 160% by 2030. The cost of treating CRC places a huge

Corresponding author
Guangxian Xu, 599040064@qq.com

financial burden on the health care system and society as a whole (*Song et al., 2020*). A variety of factors are involved in the development of CRC. The carcinogenic process of CRC includes environmental factors and genetic factors, among which the environmental factors include high-fat diet, an irregular work and rest schedule, and mental stress. Genetic factors include the complex accumulation of mutations and epigenetic regulation in key genes (*Jiang et al., 2020*). With the increasing of high mortality and recurrence rate, it is particularly important to prevent the occurrence of CRC and explore its pathogenesis. The morbidity and mortality of CRC can also be reduced through appropriate screening and surveillance (*Siegel et al., 2020*), it is necessary to explore more reliable markers to find out patients and to implement personalized treatment.

More than 70% of the human genome can be transcribed, and less than 2% of genes encode proteins, with non-coding RNA (ncRNA) accounting for the vast majority of the transcripts (*Geng, Jiang & Wu, 2018*). CircRNAs are ncRNA that have been studied extensively in recent years. Compared with linear RNAs, circRNAs do not have 5′-cap and 3′-poly (A) tail. They form a closed loop structure by covalent bond connecting the head and tail, which makes it difficult to be degraded by exonuclease, and therefore is more stable than linear RNA (*Tao et al., 2021*). CircRNA biogenesis takes place in the nucleus begin with a backsplicing reaction. Then multiple functions are displayed after circRNAs released into the cytoplasm, and the functions included circRNA could bind to miRNA and RBPs, participate in translation regulation, protein scaffold, histone modification, RNA maturation. Moreover, vesicles contains circRNAs were released from cytoplasm may be considered as a biomarker (*Chioccarelli et al., 2020*). Currently, a variety of studies have demonstrated that circRNAs can regulate gene expression in biological processes, participate in the occurrence and development of various diseases, and is involved in pathogenesis of various cancers (*Zheng et al., 2020*). In the field of reproduction, studies have shown that differentially expressed circRNAs could be used as biomarkers for sperm quality evaluation, the expression of circRNA in sperm is second only to that in brain (*Manfrevola et al., 2020*). In view of the disease, some researchers found that circFKBP8 and circMBNL1 in whole-blood had clinical diagnostic value for major depressive disorder, and circFKBP8 could be used for the antidepressive treatment (*Shi et al., 2021*). In cancer research, *Wang et al. (2021)* found that circBACH2 was up-regulated in triple-negative breast cancer. *Xu et al. (2021)* thought that hsa_circ_0003288 involved in regulation of hepatocellular carcinoma. For CRC, *Zhou et al. (2020)* proved circCAMSAP1 was significantly upregulated in CRC tissues. CircCAMSAP1 functions as an oncogene through miR-328-5p/E2F1 Axis. It can be used as a potential therapeutic target as well as a diagnostic and prognostic biomarker for CRC. And Zeng confirmed that circHIPK3 (*Zeng et al., 2018*) promoted CRC growth and metastasis though sponging miR-7, circHIPK3 promoted CRC cells proliferation, migration, invasion, and inhibited apoptosis *in vitro* and induced CRC growth and metastasis *in vivo*. Another study revealed that circRNA_0000392 (*Xu et al., 2020*) promoted tumor growth in CRC via the miR-193a-5p/PIK3R3/Akt axis in CRC cells. It has been shown that abnormal expression of circRNA can accumulate the wrong gene products, the processes will lead to the irreversible occurrence of malignant tumor formation. The mechanism might be circRNAs function as sponges for microRNAs

(miRNA) or bind to proteins (*Yu et al., 2018*) . Therefore, circRNAs may become biological indicators for cancer prediction, diagnosis and treatment.

In the present research, the circRNAs differential expression profile was analyzed using three pairs of circRNAs chip in CRCs and six potential key circRNAs were identified by qRT-PCR. In addition, ROC curve of the six circRNAs were shown. The circRNA competing endogenous RNA (ceRNA) network related to CRC was established. GO and KEGG pathway analysis were conducted. Finally, target genes according to cirReNET were analyzed by string database and a plugin app CytoHubba in cytoscape, and the hub genes were screened to show expression of the differences between colorectal cancer and adjacent normal tissues. Several core cirReAXEs were found in CRC. Having taken cirReNET into account, our finding may offer a novel insight into the molecular mechanisms of CRC, and these circRNAs may become key molecules in the diagnosis, prediction and treatment of CRC.

The aim of this study was to screen and identify novel circRNA biomarkers by circRNA chip and qRT-PCR technology. And this study supplemented the potential role of CirReNET in the development of colorectal cancer, improved our understanding of the pathogenesis of colorectal cancer. The study provided a new basis for the molecular mechanism of the occurrence and development of CRC.

## MATERIALS & METHODS

### Tissue acquisition

A total of 33 inpatients with colorectal cancer were enrolled who did not accept any form of chemotherapy, radiation and targeted therapy. They received surgical treatment in the Colorectal Surgery Department of the Affiliated Hospital of Ningxia Medical University from April 2019 to December 2020. The procedure was approved by the ethics committee and all patients signed informed consent. CRC tumor and normal tissues (tissues within 10–15 cm adjacent to the carcinoma for CRC, tissues within 5–8 cm adjacent to the carcinoma for rectal cancer) were collected and stored at −80 °C. All normal tissues were confirmed without tumor cell infiltration by pathologist after taking edge section. The clinicopathological features belong to 33 inpatients (Table 1) are shown. For circRNA chip detection, three pairs of CRC tumor and adjacent normal tissues were used, clinical characteristic were as following: tumor differentiation was moderate, no lymph node metastasis, and TNM stage (IIB). These studies involving human participants were reviewed and approved by the Ethical Committee of General Hospital of Ningxia Medical University (approval no. KYLL-2021-37).

### CircRNA chip detection

Having homogenized with TRIzol reagent (Invitrogen, Waltham, MA, USA) by the Mini-Bead-Beater-16 (Biospec, United States), Total RNA of three pairs of collected samples were then extracted used RNeasy mini-kit (Qiagen, Hilden, Germany), while simultaneous digestion was performed with DNase (Baseline Zero DNase, Epicentral, USA). NanoDrop ND-1000 spectrophotometer (Thermo Fisher Scientific, Waltham, MA, USA) was used to evaluate RNA quantity, and denaturing agarose gel electrophoresis was applied to

**Table 1  Clincopathologic characteristics of CRC patients.**

| Variables | Case, $n$(%) |
|---|---|
| Age (M ± SD, years) | 60.39 ± 10.29 |
| Gender | |
| Female | 14 |
| Male | 19 |
| Tumor location | |
| Colon | 12 |
| Rectum | 21 |
| Lymphatic invasion | |
| No | 19 |
| Yes | 14 |
| Venous invasion | |
| No | 23 |
| Yes | 10 |
| Nerve invasion | |
| No | 17 |
| Yes | 16 |
| Tumor size(cm) | |
| ≤5 | 30 |
| >5 | 3 |
| Malignant degree | |
| Low/Moderate risk | 26 |
| High risk | 7 |

assessed RNA integrity, as we described before (*Jiang et al., 2020*). After the required RNA samples were prepared, microarray hybridization was performed according to the standard protocol of ArrayStar. For enriching circular RNAs, Rnase R (Epicentre, Inc.) was used in total RNAs to remove linear RNAs. Then a random priming method (Arraystar Super RNA Labeling Kit; Arraystar) was utilized to amplify and transcribe the enriched circular RNAs into fluorescent cRNA. The fluorescent cRNAs were hybridized by the Arraystar Human circRNA Array V2 (8x15K, Arraystar). The Agilent Scanner G2505C was used to scan arrays after the slides having been washed. Acquired array images was analyzed by Agilent Feature Extraction software (version 11.0.1.1) (*Zou et al., 2020*). R software limma package was used for quantile normalization and subsequent data processing.

## Quantitative real-time PCR

SuperScript III Reverse Transcriptase (Invitrogen, Waltham, MA, USA) was used to reverse RNA into complementary DNA (cDNA). The forward (F) and reverse (R) primer sequences (Table S1) for qRT-PCR were designed by a software named CircPrimer (*Zhong et al., 2018*),  synthesized by AnHui General biosystems Co., Ltd. (Chinese). The procedure used for qRT-PCR was 95.0 °C for 2 min, and 40 circles of 95.0 °C for 5 s, 60 °C for 30 s and 95.0 °C for 15 s, 60.0 °C for 1 min and 95.0 °C for 15 s using SYBR Green PCR Master

Mix system. RNA levels were normalized to GAPDH expression and $2^{-\Delta\Delta CT}$ method was used for calculating the fold changes to determine the mRNA expression levels.

## CircRNA-miRNA-mRNA interaction prediction

These CRCs circRNA microarray results were integrated to screen commonly dysregulated circRNAs. The top three up-regulated circRNAs and the top three down-regulated circRNAs were chosen for analysis. We used the Cancer-Specific CircRNA Database (CSCD) to show the fundamental structure of circRNAs. TargetScan and miRanda databases were used to predict the interations of circRNA and miRNA. For miRNA target genes, there databases which included TargetScan, miRanda v5, and miBase were used, the candidate mRNAs exist in at least two databases simultaneously. Cytoscape software (version 3.6.1) was used to build the cirReNET for visualization.

## Gene Ontology (GO) annotation and kyoto encyclopedia of genes and genomes (KEGG) pathway analysis

GO analysis was used to construct and demonstrate meaningful annotation of gene products in multifarious organisms. Biological process (BP), cellular components (CC) and molecular function (MF) were included. KEGG covered molecular interaction and reaction networks in genes. Differentially expressed mRNAs were mapped to GO analysis and KEGG pathways in order to acquire their function information. Then the candidate mRNA in cirReNET was analyzed by GO and KEGG pathways. GO term (BP, CC and MF) and KEGG pathway were considered have statistical significance ($P < 0.05$, FDR $< 0.05$) in the results.

## Protein–protein interaction (PPI) network construction

The miRDB, miRWalk and TargetScan database were used to predict the mRNAs. For each miRNA, only these target genes predicted by three databases were retained. Target genes of the top three up-regulated circRNAs and the top three down-regulated circRNAs were analyzed separately by Search Tool for the Retrieval of Interacting Genes/Proteins (string) database to establish the PPI network. The cytoscape plugin app CytoHubba was used to rate target genes, and those with high scores were identified and listed as hub genes. Then the top 30 hub genes were entered into string to draw PPI network.

## Validation and analysis of hub genes

The GEPIA 2.0 (*Tang et al., 2017*) was used to analysis the hub genes in The Cancer Genome Atlas TCGA (COAD), the GEPIA database is a web server provided profiling and interactive analyses for cancer and normal gene expression. The top 50 hub genes were entered into GEPIA to obtain analysis results. The boxplot was formed with $|\mathrm{Log}_2\mathrm{FC}| = 1$, $P$-value $< 0.05$ and visualized with $\log_2 (\mathrm{TPM} + 1)$ for log-scale.

## Statistical analysis

The differentially expressed circRNAs in colorectal cancer and adjacent normal tissues were analyzed by paired $t$-test. All data were analyzed by SPSS19.0 statistics software. And $P$-value (0.05) was used as the cut-off value of statistical significance.

## RESULTS

### Overview of differentially expressed circRNA in the tissues of patients with CRC

As microarray assay is an efficient way for studying the biological function of RNA, the expression of circRNAs between cancer (CA) and normal control (NC) groups in the CRC were measured. The block diagram represents the normalized intensity of the two groups (Fig. S1). The variation was assessed using hierarchical clustering analysis and volcano plots. According to the principle of |fold change| >2, $P < 0.05$, 564 circRNAs were detected to be differentially expressed in the CA groups and NC groups (Figs. 1A and 1B). Among the differentially expressed circRNAs, 207 and 357 circRNAs were up-regulated and down-regulated, respectively. According to the nature of their source coding genes, the circRNAs were classified into five categories: Among the up-regulated circRNAs, antisense ($n = 4$), exonic ($n = 172$), intergenic ($n = 4$), intronic ($n = 13$), sense overlapping ($n = 14$). And the down-regulated circRNAs, antisense ($n = 4$), exonic ($n = 303$), intergenic ($n = 4$), intronic ($n = 21$), sense overlapping ($n = 25$) (Fig. 1C).

### Validation and verification by qRT-PCR assay

According to the defined threshold, 16 up-regulated circRNAs (fold change > 3 and $P < 0.05$) and 16 down-regulated circRNAs (fold change > 5 and $P < 0.05$) are shown in the cluster heatmap (Fig. 2A) and these circRNAs (Table 2) were manifested. The accuracy of circRNAs microarray data was evaluated by qRT-PCR. The top 3 most up-regulated circRNAs (hsa_circRNA_100833, hsa_circRNA_103828, hsa_circRNA_103831) and the top 3 most down-regulated circRNAs (hsa_circRNA_103752, hsa_circRNA_071106, hsa_circRNA_102293) were chosen for validation and verification via qRT-PCR assay. QRT-PCR results showed that the expression level of circRNAs in CRCs was consistent with the circRNAs chip data (Figs. 2B and 2C), manifesting the reliability and correctness of circRNAs chip data.

### Validation of differentially expressed circRNAs

To verify the expression of six significantly differential circRNAs, 33 pairs of CRCs and adjacent normal tissues were verified by qRT-PCR. The top three up-regulated circRNAs (hsa_circRNA_100833, hsa_circRNA_103828, and hsa_circRNA_103831) were dramatically up-regulated in CRC tissues (Figs. 3A–3C) ($P < 0.05$), and the top three down-regulated circRNAs (hsa_circRNA_103752, hsa_circRNA_071106 and hsa_circRNA_102293) were significantly down-regulated in CRC tissues (Figs. 3D–3F) ($P < 0.05$), compared with normal tissues. The results suggested that these circRNAs involved in the pathogenesis of CRC.

### The diagnostic value of circRNAs was evaluated by ROC curve

The ROC curve is often used to measure the ability of a certain index to identify a specific disease, and the value of AUC is an evaluation index to measure the merits of a model. The diagnostic value for CRC of six candidate circRNAs were evaluated, AUC = 0.6860, AUC = 0.8127, AUC = 0.7502, AUC = 0.9945, AUC = 0.9642, AUC

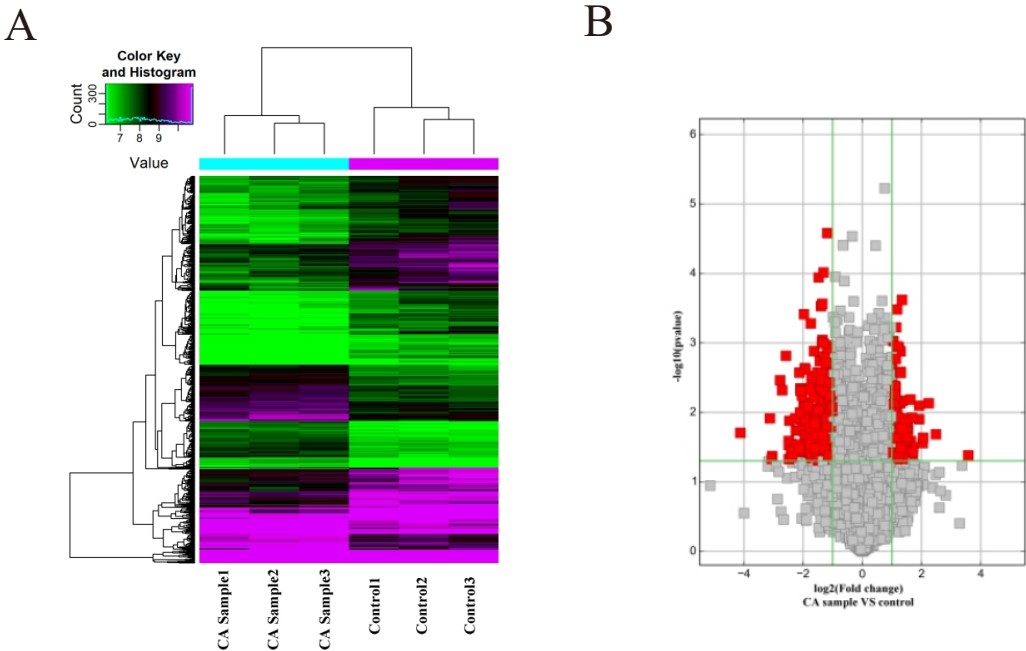

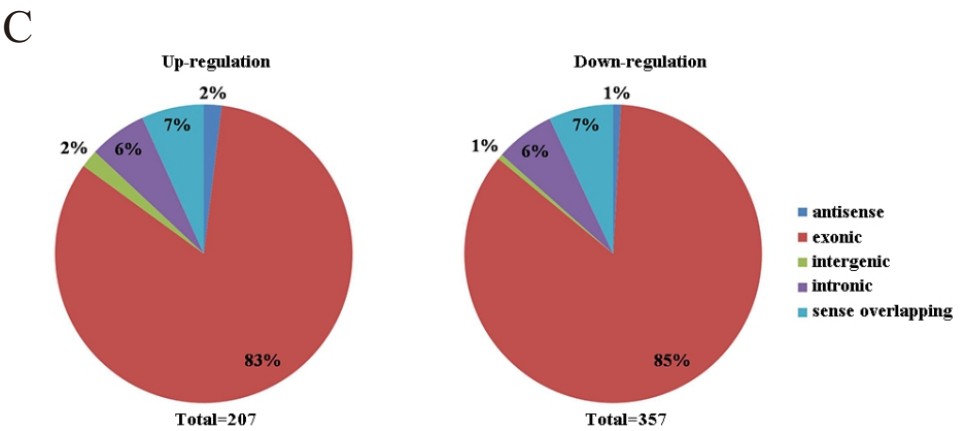

**Figure 1 Characterization of circRNA expression in CRC cells and control cells.** (A) Hierarchical clustering shows a distinguishable circRNA expression profile. (B) Volcano plot of the differentially expressed circRNAs. (C) The origin of up-regulated and down-regulated circRNAs. The red squares represent differentially expressed circRNAs with statistical significance.

= 0.9486 for hsa_circRNA_100833, hsa_circRNA_103828, hsa_circRNA_103831 and hsa_circRNA_103752, hsa_circRNA_071106, hsa_circRNA_102293, respectively (Fig. 4) ($P < 0.05$). The above data indicate that the six circRNAs have high diagnostic efficiency and can be used as indicators for the prediction and diagnosis of CRC disease.

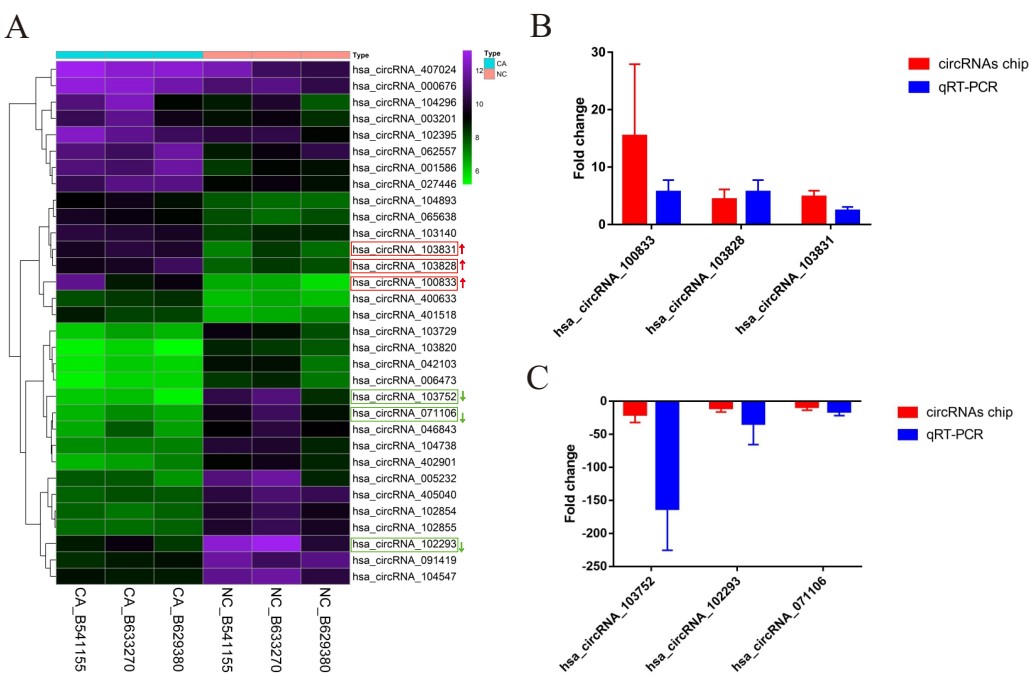

**Figure 2** **The cluster heatmap of circRNAs chip and histogram of Arraystar human circRNAs chip versus quantitative real-time PCR comparison.** (A) Chip analysis of the top sixteen most up-regulated and down-regulated circRNAs. (B) Arraystar human circRNAs chip versus qRT-PCR.GAPDH was used to normalize for measuring circRNA expression levels. NC, normal tissues. CA, cancer tissues.

## The construction of CircRNA-miRNA-mRNA regulatory network (cirReNET)

In order to understand the basic structure of the six candidate circRNAs, we used CSCD to predict the fundamental structural modes of the candidate circRNAs (Fig. S2), we can clearly grasp the MRE (microRNA response element), RBP (RNA binding protein) and ORF (open reading frame) structural regions. To further understand the binding function of circRNAs with miRNAs, TargetScan and miRanda databases were used to building circRNA-miRNA interaction network, and the top five targeted miRNAs of there up-regulated and there down-regulated candidate circRNAs (Fig. 5) were shown. According to mechanism circRNAs function as sponges for miRNA, highest context score percentile miRNA were selected to display the potential and detailed sites of circRNA-miRNA interaction (Fig. S3). Then, the cirReNET contain six circRNAs, 19 miRNAs, and 210 mRNAs, was visualized via Cytoscape software (version 3.6.1) (Fig. 6). This ceRNA network provides a comprehensive view of the interactions with circRNAs, miRNAs and mRNAs in the pathogenesis of CRC.

## GO and KEGG pathway analysis

Functional enrichment analysis was used to explore corresponding biological functions. The top 10 enrichment scores for 3 up-regulated candidate circRNAs are listed (Figs. 7A–7C). The GO analysis indicated that the term about biological process terms (BP) with the

**Table 2  The top 16 significantly up-regulated and top 16 significantly down-regulated circRNAs.**

| circRNA | Style | *P*-value | FC (abs) | chrom | strand | circRNA_type | GeneSymbol |
|---|---|---|---|---|---|---|---|
| hsa_circRNA_100833 | Up | 0.041404417 | 11.9693427 | chr11 | + | Exonic | FADS2 |
| hsa_circRNA_001586 | Up | 0.020644494 | 5.6125946 | chr6 | − | Sense overlapping | HIST1H3D |
| hsa_circRNA_103831 | Up | 0.007338473 | 4.7409963 | chr5 | − | Exonic | HMGCS1 |
| hsa_circRNA_103828 | Up | 0.023048044 | 4.1522142 | chr5 | − | Exonic | HMGCS1 |
| hsa_circRNA_104296 | Up | 0.02754745 | 4.1136695 | chr7 | − | Exonic | RNF216 |
| hsa_circRNA_400633 | Up | 0.00796824 | 3.7916743 | chr10 | + | Exonic | SCD |
| hsa_circRNA_027446 | Up | 0.01251238 | 3.7891116 | chr12 | + | Exonic | HMGA2 |
| hsa_circRNA_407024 | Up | 0.031582655 | 3.2868653 | chr7 | − | Exonic | ZNF767P |
| hsa_circRNA_401518 | Up | 0.037198732 | 3.2686815 | chr16 | − | Exonic | TRAP1 |
| hsa_circRNA_062557 | Up | 0.040687681 | 3.256663 | chr22 | − | Exonic | CHCHD10 |
| hsa_circRNA_102395 | Up | 0.041155184 | 3.2070677 | chr19 | + | Exonic | PTBP1 |
| hsa_circRNA_104893 | Up | 0.011917176 | 3.1960212 | chr9 | + | Exonic | PAPPA |
| hsa_circRNA_065638 | Up | 0.025681377 | 3.1094732 | chr3 | − | Exonic | GPX1 |
| hsa_circRNA_003201 | Up | 0.026106899 | 3.1040179 | chr4 | + | Exonic | TBC1D14 |
| hsa_circRNA_000676 | Up | 0.006494205 | 3.0807421 | chr22 | + | Sense overlapping | L3MBTL2 |
| hsa_circRNA_103140 | Up | 0.042213331 | 3.0223647 | chr21 | + | Exonic | PDXK |
| hsa_circRNA_103752 | Down | 0.019673168 | 17.4354314 | chr4 | − | Exonic | LRBA |
| hsa_circRNA_071106 | Down | 0.012172022 | 8.7288597 | chr4 | + | Exonic | ARHGAP10 |
| hsa_circRNA_102293 | Down | 0.046702335 | 8.3722472 | chr18 | + | Exonic | MTCL1 |
| hsa_circRNA_005232 | Down | 0.042405228 | 8.2508771 | chr2 | − | Exonic | SLC8A1 |
| hsa_circRNA_405040 | Down | 0.003433548 | 6.8606691 | chr12 | + | Exonic | PLXNC1 |
| hsa_circRNA_103820 | Down | 0.004792873 | 6.5462592 | chr5 | − | Exonic | LIFR |
| hsa_circRNA_046843 | Down | 0.001527746 | 5.9829024 | chr18 | + | Exonic | ANKRD12 |
| hsa_circRNA_042103 | Down | 0.04696831 | 5.6131975 | chr17 | + | Exonic | MYOCD |
| hsa_circRNA_102854 | Down | 0.025715693 | 5.5996816 | chr2 | + | Exonic | PDK1 |
| hsa_circRNA_091419 | Down | 0.029347044 | 5.5994999 | chrX | − | Exonic | RPL39 |
| hsa_circRNA_102855 | Down | 0.013142528 | 5.5183225 | chr2 | + | Exonic | PDK1 |
| hsa_circRNA_104738 | Down | 0.035917372 | 5.3141209 | chr9 | − | Exonic | BNC2 |
| hsa_circRNA_402901 | Down | 0.035727197 | 5.264933 | chr3 | − | Exonic | EIF4E3 |
| hsa_circRNA_104547 | Down | 0.036827804 | 5.2491912 | chr7 | − | Exonic | ESYT2 |
| hsa_circRNA_006473 | Down | 0.028144483 | 5.2429886 | chr4 | + | Exonic | ARHGAP10 |
| hsa_circRNA_103729 | Down | 0.044074879 | 5.2027097 | chr4 | − | Exonic | PDE5A |

highest enrichment score was about cellular metabolic process (GO: 0031323). The term about cellular component terms (CC) was cytoplasm (GO: 0005737) and nitric-oxide synthase binding (GO: 0050998) for molecular function terms (MF). In addition, the top 10 enriched KEGG pathways are listed (Fig. 7D). The pyrimidine metabolism signaling pathway has the highest enrichment score in the KEGG pathway.

In contrast, the top 10 enrichment scores for three downregulated candidate circRNAs are listed (Figs. 7E–7G). The term were related with cell cycle (GO: 0007049) and glandular epithelial cell differentiation (GO: 0002067) for BP. The term about CC with the highest enrichment score was banded collagen fibril (GO: 0098643) and the term with the highest enrichment score was protein binding (GO: 0005515) for MF. Respectively, the DNA

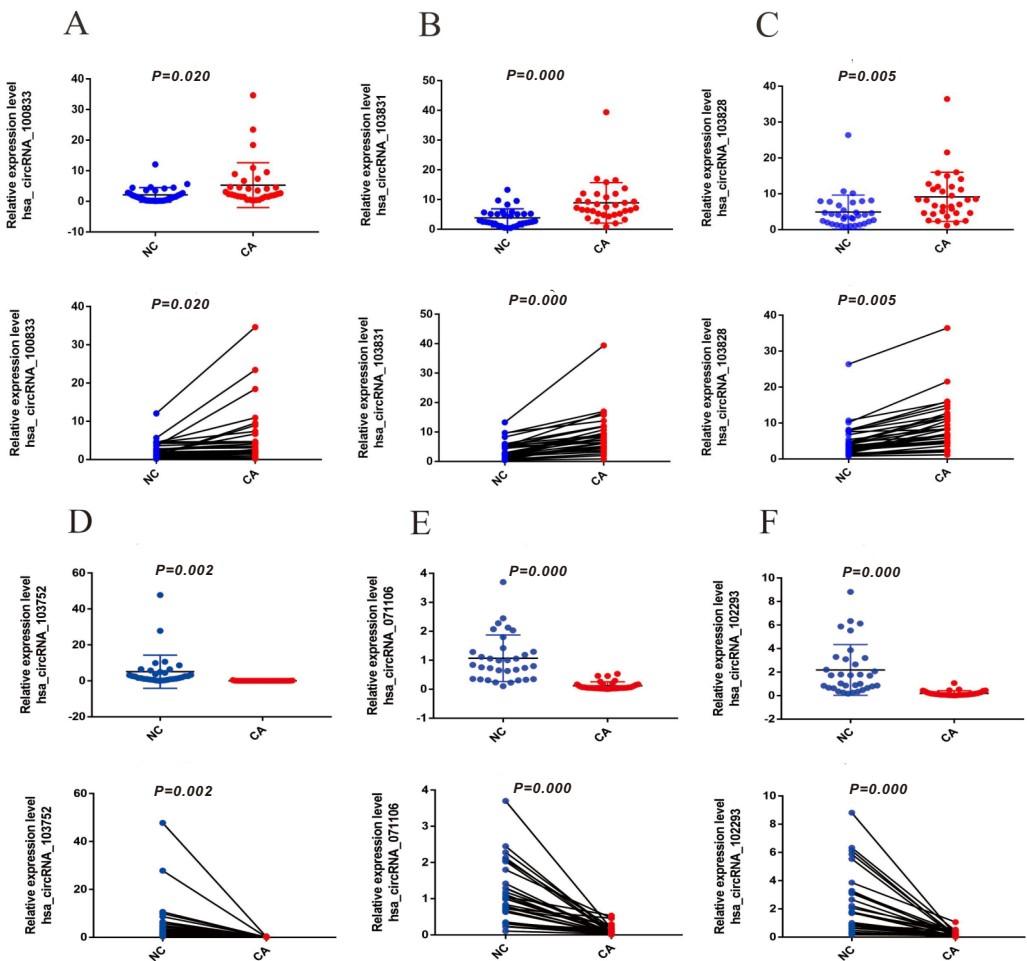

**Figure 3** **The expression of up-regulated and down-regulated circRNAs of 33 CRC tissues (A) hsa_circRNA_100833 (B) hsa_circRNA_103831 (C) hsa_circRNA_103828 (D) has_circRNA_103752 (E) hsa_circRNA_071106 (F) hsa_circRNA_102293.** GAPDH was used to normalize for measuring circRNA expression levels. $P < 0.05$ indicated a difference in comparison between the two groups, and $P < 0.001$ indicated a significant difference.

replication has the highest enrichment score in the KEGG pathway, and the top 10 enriched KEGG pathways are listed (Fig. 7H).

## Construction of PPI network

To understand more about the core interactions among the up-regulated and down-regulated target genes in cirReNET, we used the miRDB, miRWalk and TargetScan databases to predict target genes for the five miRNAs for each circRNA according to ceRNA mechanism. Only these genes predicted by three databases were retained. Then functional genes were filtered by string and the PPI networks were visualized. In addition, plugin app CytoHubba in Cytoscape was used to rate target genes according to Maximal Clique Centrality (MCC). Then the top 30 hub genes were entered into string to draw PPI network (Fig. 8), and the top 50 hub genes were listed (Table S2).

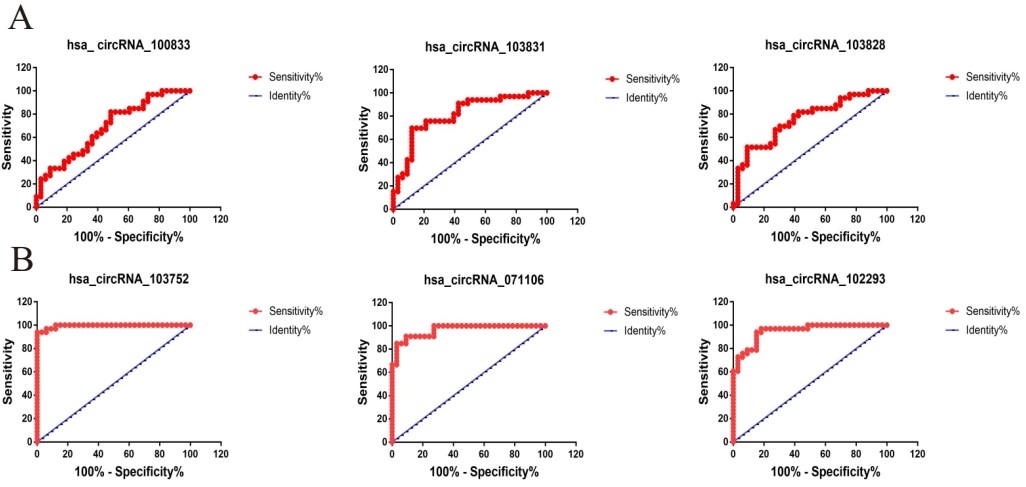

**Figure 4  The ROC curves of three up-regulated and three down-regulated circRNAs in CRC.** (A) The diagnostic value of three up-regulated circRNAs presented by ROC curve. (B) The diagnostic value of three down-regulated circRNAs presented by ROC curve.

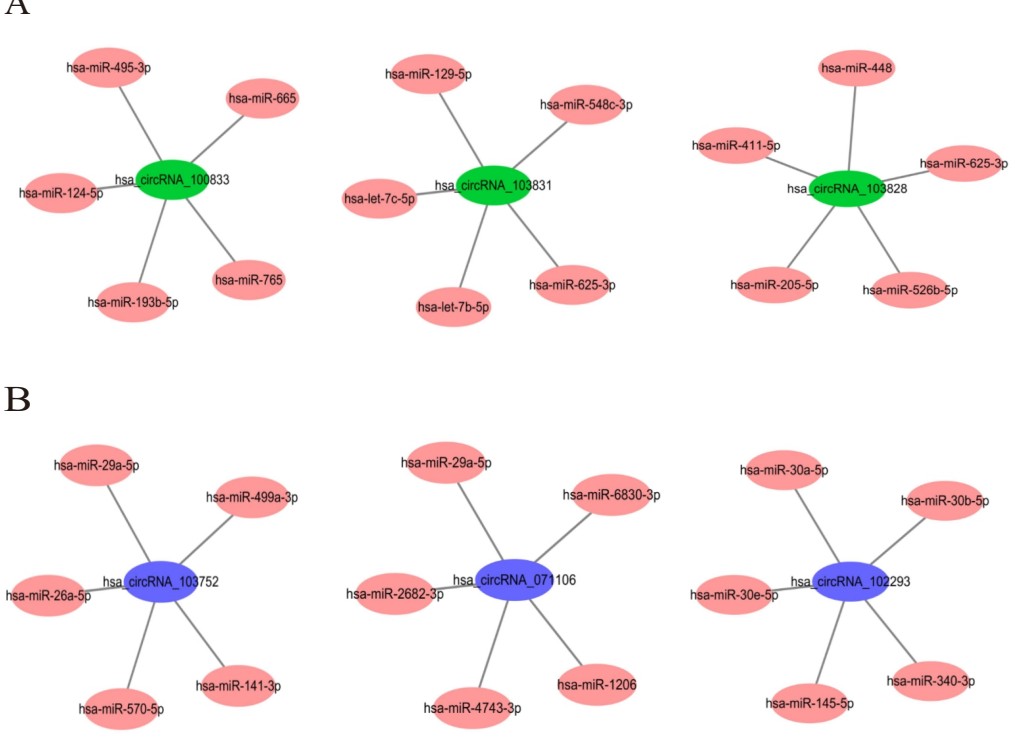

**Figure 5  The top five targeted miRNAs for circRNA-miRNA regulatory network.** (A) Up-regulated circRNAs. (B) Down-regulated circRNAs.

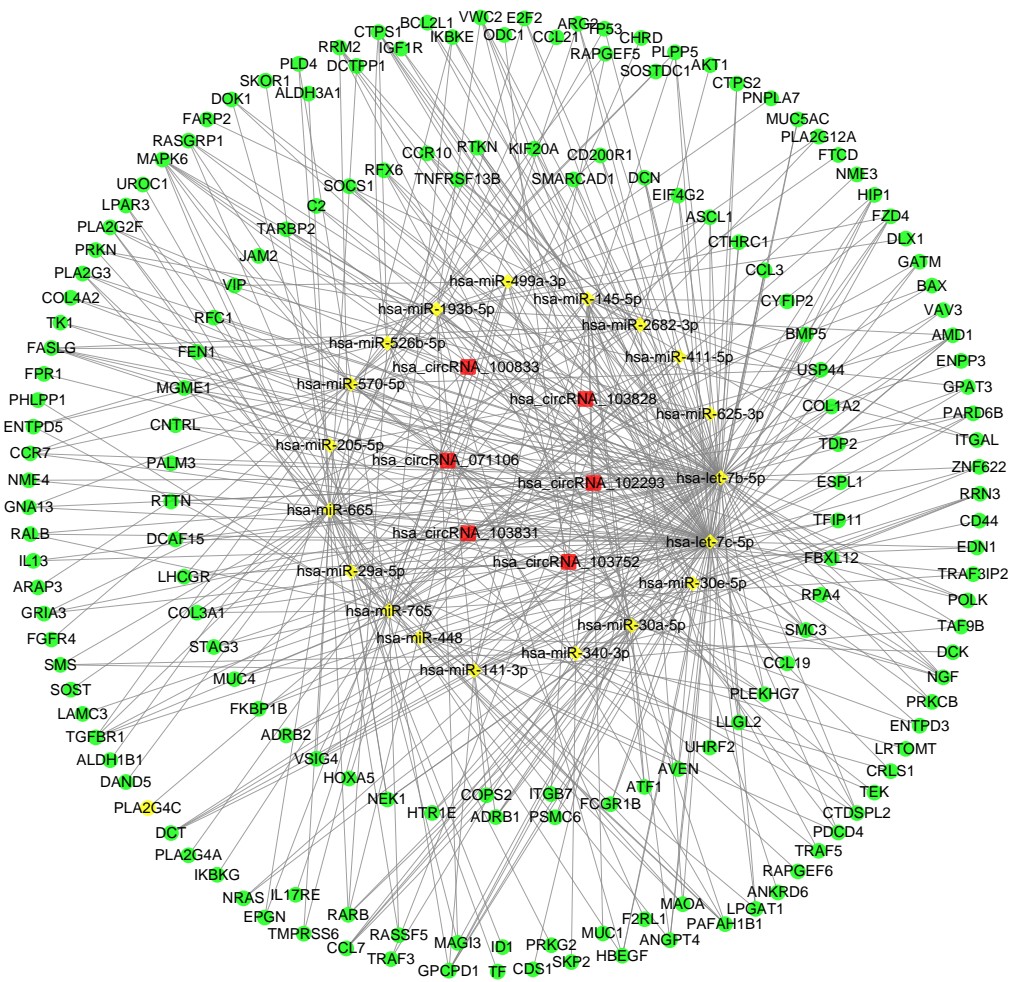

**Figure 6** **CircRNA-miRNA-mRNA regulatory network (cirReNET) in the CRC tissues.** The cirReNET included six circRNAs, 19 miRNAs and 210 mRNAs. CircRNAs are represented in red and presented as square, miRNAs are represented in yellow and presented as diamond, and mRNA are represented in green and presented as circle.

## Identification of hub genes expression

To understand more information about the expression of predicted target genes in colorectal cancer and adjacent normal tissues, GEPIA was used to calculate and show the expression of hub genes in TCGA (COAD and READ). We chose the top 50 hub genes to enter into GEPIA to screen (Fig. 9 and Fig. S4), and the 150 hub genes in the upregulated circRNA-downregulated miRNA-upregulated mRNA (UcDiUm-RNA) network, there was higher expression for 12 hub genes (CCL4, APLN, FBXO22, SH3KBP1, CKAP4, PDIA6, RCN1, TP53, CCND1, EZH2, E2F2, CASP3) in the CRC group as compared to normal tissues group. And the 150 hub genes in the downregulated circRNA-upregulated miRNA-downregulated mRNA (DcUiDm-RNA) network, 10 hub genes (RNF217, KLHL5, UNKL, PTGER3, FBXO32, TRIM9, KCTD7, KLHL42, SRSF11, RBM5)were lower expression in

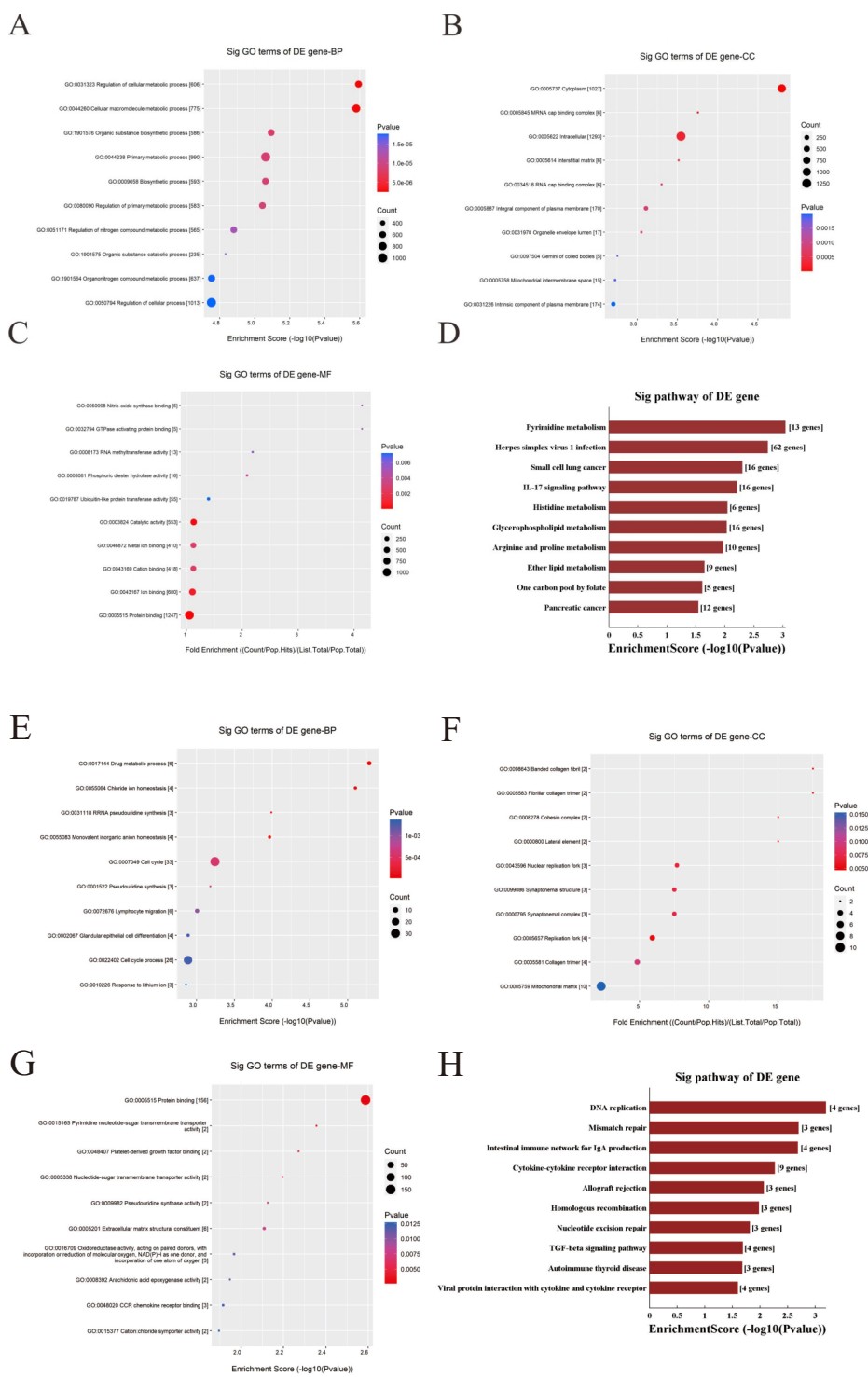

**Figure 7   GO and KEGG pathway analysis for three up-regulated circRNAs and three down-regulated circRNAs.** (A–C & E–G) GO annotation of targeted genes with the top 10 enrichment scores for BP, CC, and MF, respectively. (D&H) The top 10 enriched KEGG pathways. Enrichment score was calculated as −log10 (*P*-value). (A–D) Up-regulated circRNAs. (E–G) Down-regulated circRNAs. GO, gene ontology. KEGG, Kyoto Encyclopedia of Genes and Genomes. BP, biological process. CC, cellular component. MF, molecular function.

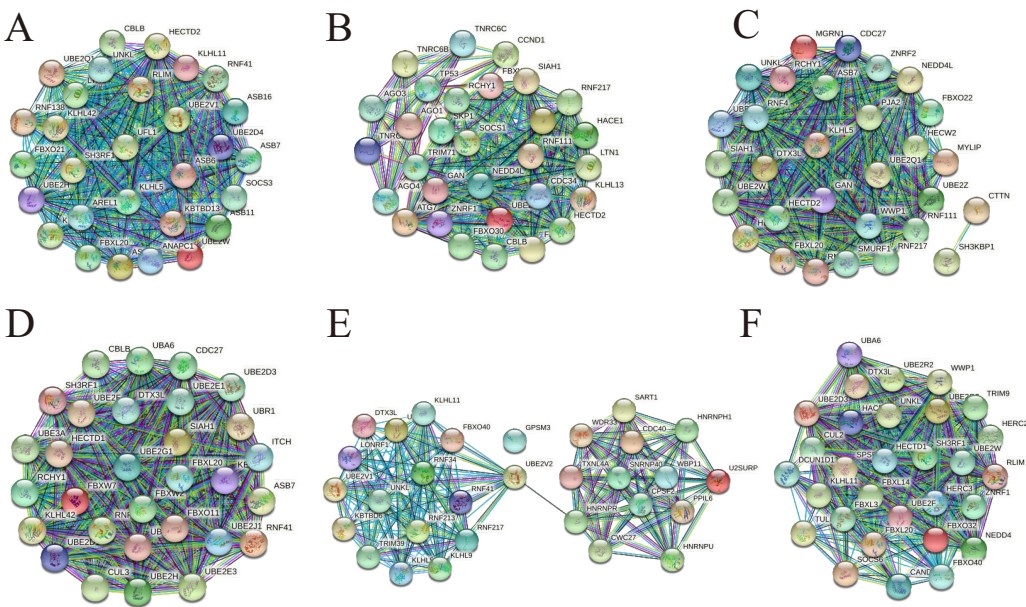

**Figure 8** **PPI networks consisting of hub genes belong to three up-regulated circRNAs and three down-regulated circRNAs.** (A) The top 30 hub genes belong to hsa_circRNA_100833. (B) The top 30 hub genes belong to hsa_circRNA_103831. (C) The top 30 hub genes belong to hsa_circRNA_103828. (D) The top 30 hub genes belong to hsa_circRNA_103752. (E) The top 30 hub genes belong to hsa_circRNA_071106. (F) The top 30 hub genes belong to hsa_circRNA_102293.

the CRC group in the DcUiDm-RNA network. In addition, we may have predicted and discovered several critical cirReAXEs that play important roles in CRC (Fig. 10), the next step is further experimental confirmation.

## DISCUSSION

CRC accounts for a significant proportion of cancer deaths and is one of the most commonly diagnosed malignancies (*Song et al., 2020*). CircRNAs are newly discovered and ubiquitous endogenous ncRNAs (*Lu et al., 2018*). Multiple studies have shown that circRNAs function as sponges for miRNAs and regulate the corresponding target genes expression to affect diseases. *Han et al. (2021b)* found that the up-regulated circRNA hsa_circ_0071036 sponge miR-489 in pancreatic cancer can promote the oncogenic process. *Liu et al. (2021)* thought that circMTO1 suppressed proliferation and metastasis of osteosarcoma through miR-630/KLF6 axis. According to the current studies, abnormally expressed circRNAs participate in the process of CRC carcinogenesis and regulate the expression of related genes and proteins. *Li et al. (2018b)* have confirmed that circRNA_101951 regulated EMT pathway could promote migration and invasion of CRC cells (*Li, Pei & Cao, 2020b*). Another study revealed circDDX17 played as tumor suppressor in CRC. And a report has shown that circ_0115744 sponge miR-144 to regulate the metastasis of CRC (*Ma, Lv & Xing, 2021*). *Han et al. (2021a)* proved that circNSUN2/miR-296-5p/STAT3 axe was under control of Aloperine to inhibit proliferation and promote apoptosis of CRC cells. *Zhi et al. (2021)*

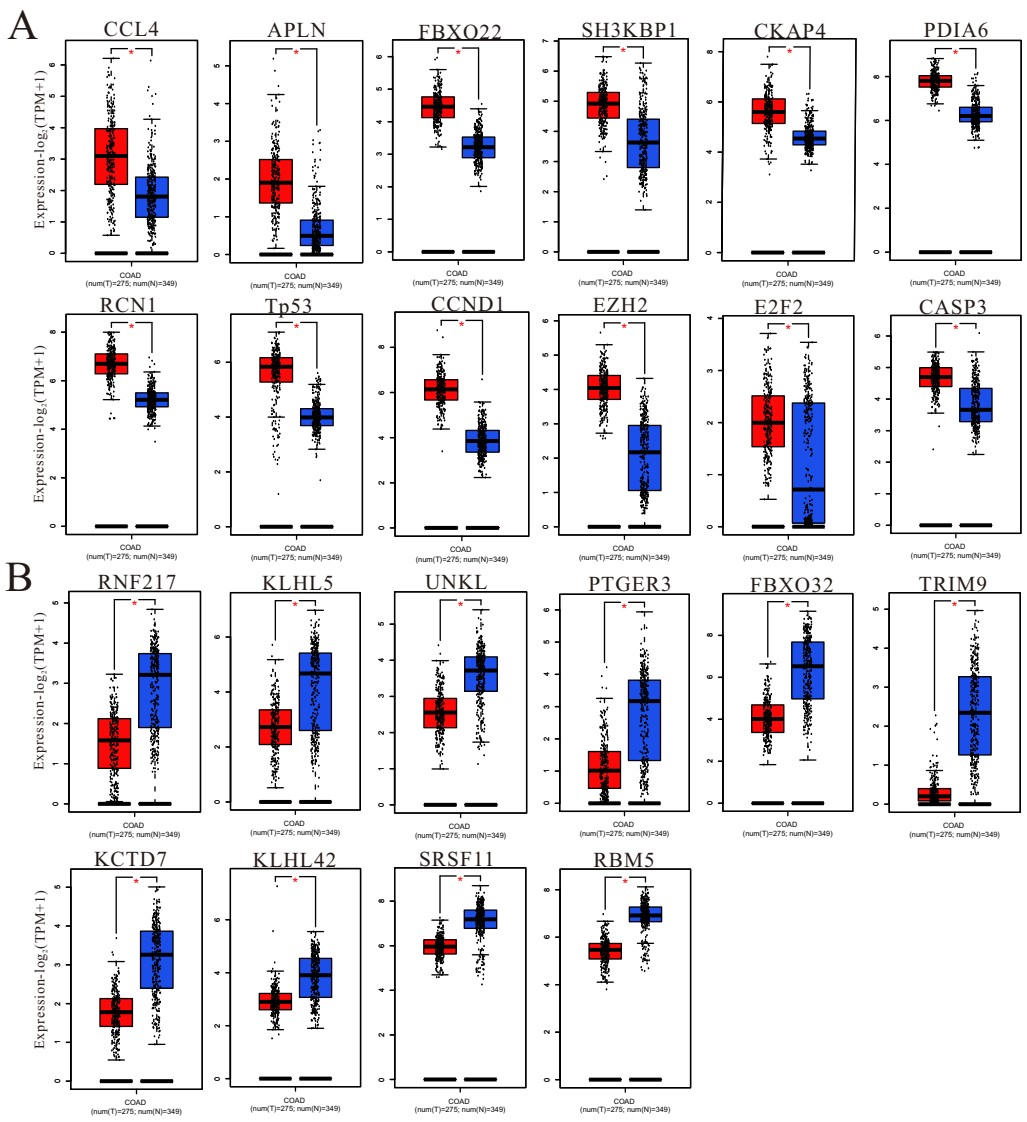

**Figure 9** **Hub gene expression.** (A) Up-regulated hub genes via GEPIA analysis in cancer tissues. (B) Down-regulated hub genes via GEPIA analysis in cancer tissues. An asterisk (*) represent $P < 0.05$, $P < 0.05$ indicated a difference in comparison between the two groups.

found that circ102049 was identified as a core member by circRNA profiling in colorectal liver metastasis. In general, current studies on circRNAs in CRC mainly focus on the regulatory mechanism of miRNA sponge.

The pathogenesis of CRC is generally related to the following two reasons (*Paschke et al., 2018*): the incidence of CRC is related to chromosomal instability (CIN). The activation of oncogenes such as KRAS or the occurrence of a series of events such as adenomatous polyposis coli gene (APC), DCC and p53 inactivation lead to the gradual evolution of normal intestinal mucosa-adenomato-carcinoma. Juvenile stem cell mutations mismatch repair genes (MMR) lead to hereditary nonpolyposis colorectal cancer, or somatic

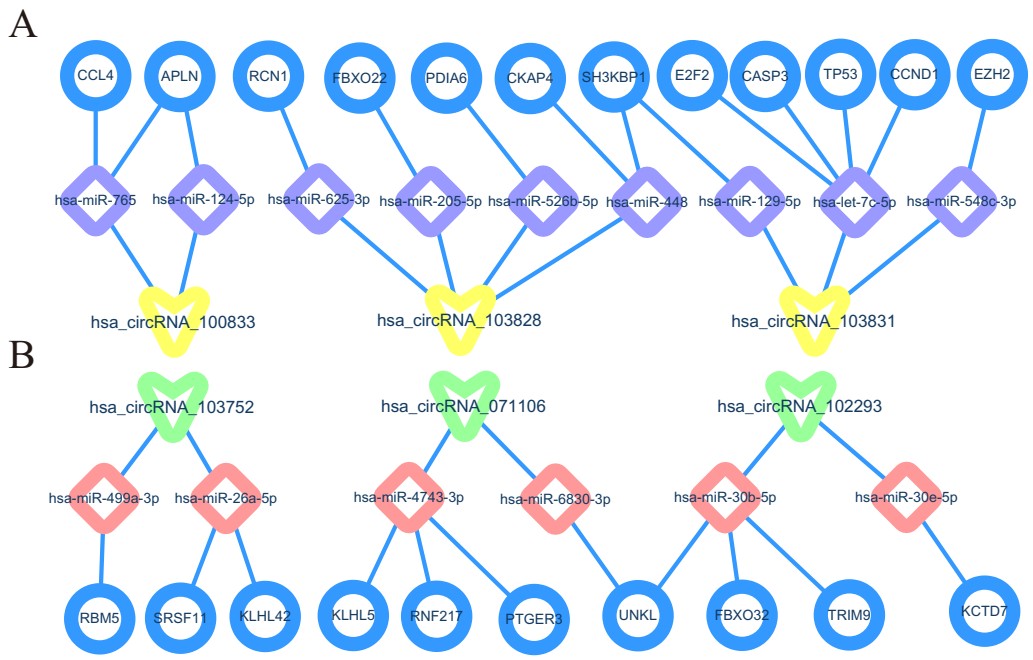

**Figure 10   Prediction of core circRNA-miRNA-mRNA axes (cirReAXEs) in the CRC.** (A) Core networks for three up-regulated circRNAs. (B) Core networks for three down-regulated circRNAs.

microsatellite instability (MSI) can eventually lead to tumor formation. The pathogenesis is not exactly the same in colon and rectal cancer, colon cancer is more affected by family inheritance than rectal cancer, rectal cancer has a high incidence of CIN, low incidence of MSI. The main pathogenesis of rectal cancer is the APC/β-catenin pathway, and the mutation frequency of KRAS and other genes in rectal cancer is higher than that in colon cancer, and the APC restrictive mode is more common. P53 gene can be highly expressed in rectal cancer, and COX2 can be highly expressed in 90% of rectal cancer and 20% of colon cancer (*Frattini et al., 2004*). Therefore, for circRNA, the differential expression of non-coding RNA in colon cancer and rectal cancer, except for genes with the same expression trend, may also show corresponding differences.

To investigate the differential expression of circRNAs in CRC, circRNA chip detection was chosen to profile and demonstrate three pairs tissues belong to patients with CRC in the present study. According to the principle of |fold change| >2, $P < 0.05$, there are 564 differentially expressed circRNAs were detected between the CA group and NC group. Among the differentially expressed circRNAs, 207 up-regulated and 357 down-regulated circRNAs in CRCs were screened, respectively. Then, three up-regulated circRNAs (hsa_circRNA_100833, hsa_circRNA_103828, hsa_circRNA_103831) and three down-regulated circRNAs (hsa_circRNA_103752, hsa_circRNA_071106, hsa_circRNA_102293) were identified to confirm consistency with the expression of the chip results. Then, 33 pairs of CRC tissues and adjacent normal tissues were collected to verify and validate the expression of these six circRNAs in our research. The analysis results show that high diagnostic values were involved in all of these circRNAs.

For the cirReNET we built, there were many targeted genes involved in the mechanism of tumor genesis, such as AKT1 (*Liu et al., 2020*), ErbB2 (*Li et al., 2018a*), ITGB7 (*Sun et al., 2020*), BAX, MAPK, FBXL12 (*Nita et al., 2016*), CCL7 (*Zhang et al., 2020b*) and so on. These genes also play very important roles in cell proliferation, apoptosis, movement, T-cell differentiation, adhesion, and polarity formation. At present, a large number of circRNA studies are still in progress, but the carcinogenic or anticancer effects of many miRNAs in CRC are clear. For example, in Fig. 5, hsa_circRNA_100833 is highly expressed in CRC in our experiment. It is predicted that the miRNAs targeted by hsa_circRNA_100833 are low expressed and can inhibit the proliferation and growth of CRC cells. *Ma & Deng (2021)* confirmed that miR-665 plays a role in inhibiting the proliferation, migration and invasion of CRC cells. MiR-448 (*Zhang et al., 2019*) has also been confirmed to play a tumor suppressive role by promoting cell apoptosis. Similarly, for circRNA with low expression in CRC, such as hsa_circRNA_071106, its predicted miRNAs should play a carcinogenic role. It has been reported that overexpression of miR-29a-5p (*Wang et al., 2020*) in colorectal cancer cells can amplify inflammatory effects and promote CRC progression through the STAT3 signaling pathway. The above results confirm the reliability of our prediction.

The functions of these target genes were then assessed and analyzed with the GO analysis. For target genes corresponding to the three up-regulated-circRNAs, GO enrichment analysis manifested many crucial biological processes included in these genes, such as cellular metabolic process, biosynthetic process and so on. Meanwhile KEGG pathway analysis provided a deep insight of the pathogenesis of CRC. KEGG pathway analysis was performed to show some vital pathways, include IL-7 signaling pathway, Glycerophospholipid metabolism, Histidine metabolism etc. For other three down-regulated-circRNAs, GO enrichment analysis revealed target genes involved in regulating cell cycle, glandular epithelial cell differentiation. KEGG pathway analysis showed that DNA replication, mismatch repair and TGF-beta signaling pathway might be the mechanism of the development of CRC.

In order to further understand the core interactions among the up-regulated and down-regulated target genes in cirReNET, the string was used to filter functional genes and the PPI networks were visualized. According to cirReNET, thousands of mRNA are indirect target genes of circRNA. The GEPIA database can provide fast and comprehensive annotation of gene expression in TCGA (COAD). The Hub genes presented in this paper are mRNAs that are consistent with circRNA expression in CRC and verified via TCGA database. GEPIA not only screens genes predicted by bioinformatics, but also validates the results of bioinformatics analysis. The application of GEPIA provides further evidence that our predicted cirReAXEs is involved in the pathogenesis of CRC.

Differentially expressed up-regulated and down-regulated hub genes were shown by GEPIA, we may have found several core cirReAXEs in CRCs. such as has_circRNA_100833-hsa-miR-124-5p-APLN axe, hsa_circRNA_103831-hsa-let-7c-5p-TP53 axe, hsa_circRNA_071106-hsa-miR-6830-3p-UNKL. These core cirReAXEs may participate in major regulation in the pathogenesis of CRC. In Fig. 10, we can observe that one circRNA can target several miRNAs, and one miRNA can target several mRNAs, the same mRNA was regulated by two different miRNAs. In addition, one circRNA can

target two miRNA members coming from the same family, such as hsa-circRNA-102293 target same miRNA family (miR-30b-5p and miR-30e−5p). This trend also appears in the regulation of miRNA to mRNA. That is because miRNAs typically consist of about 20 nucleotides, and pairing with the base of the target mRNA or circRNA depends on 7 or 8 nucleotides at the 5′ end of miRNA (called the "seed region") (*Goodall & Wickramasinghe, 2021*). Furthermore, miRNAs with highly homologous sequences (especially "seed region") are classified as a miRNA family, and members of the same miRNA family have similar functions. Therefore, we can understand that a single miRNA can have multiple mRNA targets, and a single mRNA can be regulated by multiple miRNAs, and the same circRNA or mRNA can target several miRNAs in the same family.

There are several limitations in our study. Firstly, only 33 patients were enrolled, the sample size is relatively small. We applied Gpower software to select post-hoc analysis for power analysis, and the power value was 0.9998, and the result greater than 0.8 proved that the sample size was sufficient to explain the problem. Secondly, we only conducted a network based on identified three up-regulated circRNAs and three down-regulated circRNAs in our work, other circRNAs may be also involved in regulation in mediating the development of CRC, but we have not explored. Furthermore, our paper is a simple descriptive analysis of the expression of circRNAs in CRC, but the function and mechanism of each circRNA in CRC have not been verified and discussed in detail.

In our study (Fig. S5), we first conducted circRNA chip analysis on the samples of three CRC patients, and obtained a total of 564 differentially expressed circRNAs through the intersection of the chip analysis of three patients and corresponding restriction rules. Three top up-regulated and down-regulated circRNAs were verified using qRT-PCR (33 clinical samples). Six circRNAs were evaluated by ROC curve. These six circRNAs have potential function as CRC markers and can be used as targets for prediction, screening and treatment. Next, we predicted cirReNET through bioinformatics analysis. GO and KEGG analyses were performed for the mRNAs indirectly targeted by these six circRNAs, and the target genes were ranked by CytoHubba, a plug-in of Cytoscape software. The top 30 genes were taken from the mRNA indirectly targeted by each circRNA to draw the PPI network diagram. Meanwhile, the top 50 target genes were screened and verified by GEPIA database, and multiple cirReAXEs were further obtained. The target genes of cirReAXEs were verified by TCGA database to be more reliable. Therefore, in our study, we confirmed that these 6 circRNAs are potential targets or biomarkers for the diagnosis and treatment of CRC. In addition, several key cirReAXEs were obtained through corresponding bioinformatics analysis and database verification. In our future studies, we will study these circRNAs in cell and animal models by inhibiting or overexpressing circRNA, and verify the mechanism of cirReAXEs proposed by us. In the present study, we identified six key circRNAs which can be used as biomarkers for prediction, treatment and diagnosis of CRC. Potential cirReNET mechanisms in CRC are predicted by bioinformatics, and further studies are urgently to verify molecular mechanisms.Our study provides new insights into the mining of new circRNAs as CRC biomarkers and their potential mechanisms in CRC.

## CONCLUSIONS

Six circRNAs with potential predictive, diagnostic and therapeutic value were obtained by circRNA analysis, qRT-PCR validation and ROC curve analysis. Several important cirReAXEs that may be involved in the development of CRC were obtained by bioinformatics analysis as well as by the screening and verification of TCGA databases.

## ACKNOWLEDGEMENTS

We thank Dr. Dong Zhang and Department of Colorectal Surgery for providing the specimens in General Hospital of Ningxia Medical University.

### Funding

This research was funded by the National Natural Science Fund of China (No. 81860355) and Ningxia Key Research and Development Program (No: 2020BFG02017) and Discipline construction project of Guangdong Medical University and Ningxia Key Research and Development Program (2018BEG02007). The funders had no role in study design, data collection and analysis, decision to publish, or preparation of the manuscript.

### Grant Disclosures

The following grant information was disclosed by the authors:
National Natural Science Fund of China: 81860355.
Ningxia Key Research and Development Program: 2020BFG02017.
Discipline construction project of Guangdong Medical University and Ningxia Key Research and Development Program: 2018BEG02007.

### Competing Interests

The authors declare there are no competing interests.

### Author Contributions

- Li yuan Liu conceived and designed the experiments, performed the experiments, analyzed the data, prepared figures and/or tables, authored or reviewed drafts of the paper, and approved the final draft.
- Dan Jiang conceived and designed the experiments, analyzed the data, prepared figures and/or tables, authored or reviewed drafts of the paper, and approved the final draft.
- Yuliang Qu conceived and designed the experiments, analyzed the data, prepared figures and/or tables, authored or reviewed drafts of the paper, and approved the final draft.
- Hongxia Wang performed the experiments, authored or reviewed drafts of the paper, and approved the final draft.
- Yanting Zhang performed the experiments, authored or reviewed drafts of the paper, and approved the final draft.
- Shaoqi Yang performed the experiments, authored or reviewed drafts of the paper, and approved the final draft.

- Xiaoliang Xie performed the experiments, authored or reviewed drafts of the paper, and approved the final draft.
- Shan Wu performed the experiments, authored or reviewed drafts of the paper, and approved the final draft.
- Haijin Zhou performed the experiments, authored or reviewed drafts of the paper, and approved the final draft.
- Guangxian Xu conceived and designed the experiments, performed the experiments, analyzed the data, prepared figures and/or tables, authored or reviewed drafts of the paper, and approved the final draft.

## Human Ethics

The following information was supplied relating to ethical approvals (i.e., approving body and any reference numbers):

These studies involving human participants were reviewed and approved by the Ethical Committee of General Hospital of Ningxia Medical University (approval no. KYLL-2021-37).

## Data Availability

The circRNA microarray data is available at GEO: GSE197991 and in the Supplemental File.

## Supplemental Information

Supplemental information for this article can be found online at http://dx.doi.org/10.7717/peerj.13420#supplemental-information.

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
