# Peer review of "Potential and functional prediction of six circular RNAs as diagnostic markers for colorectal cancer"

_PeerJ, doi:10.7717/peerj.13420_

## Round 0.1 · original submission · Major Revisions

All of the issues raised by the reviewers should be addressed thoroughly.

Besides, the following points need explanation:

1. Details of the bioinformatics analysis of the manuscript are not clear enough. The steps of the analysis should be stated. How the COAD data is used? How was this analysis planned?

2. The patient samples included in the study consists of 12 colon and 21 rectum cancer cases. However, in the GEPIA analysis on TCGA data only COAD (colon adenocarcinoma) dataset is included. When 64% of the patient samples used were rectum tumour, why the authors didn't include READ (rectum adenocarcinoma) dataset in their analyses?

3. The discussion of GEPIA analysis is insufficient. It needs to be thoroughly discussed and explained how this analysis help to prove the hypothesis of the study.

4. The candidate biomarkers suggested in the study were not tested in COAD and READ datasets. However, that would strengthen the findings.

·

Basic reporting

The authors were used clear and unambiguous English throughout the manuscript. However, some abbreviations were not written explicitly where they are first used.
The article includes sufficent introduction that shows how the work fits into the broader field of knowledge and current literature was referenced.
There are too many figures in the manuscript, the number of figures should be reduced.

Experimental design

The subject of the manuscript fits with the aims and scope of the journal.

Validity of the findings

In the literature there are several studies showing the relation between the role of circRNAs and colorectal cancer pathogenesis. From this point of view, although the originality of the article is low, it is thought that it may contribute to the literature since the circRNAs that are found in the studied patient group are different from the literature. I believe that this manuscript can have a potential for publication in PeerJ after considering major concerns.

1. In the materials and methods parts, CRC tumor and paracancerous tissues (as tissue within 10-15 cm adjacent to the carcinoma) were used in order to make a comparison between the differential expression profile of circRNAs of tumor and normal tissues. However, paracancerous tissue is refered as the tissue less than 2 cm away from the tumor edge while normal tissue was defined as the tissue more than 2 cm away from the tumor edge. In this study, it is not clear that whether the control tissue is paracancerous or normal tissue. Authors did not indicate any pathological data regarding that control tissue is completely free of tumor cells.
2. Colorectal cancer is a type of cancer which originates from colon or rectum cells. However, still many apparent differences exist in terms of genetics, anatomy and treatment response between colon cancer and rectal cancer. In this manuscript out of 33 cases 12 of them have colon and 21 of them have rectum tumor. In the results and discussion parts circRNAs differential expression patterns should be considered between these groups also.
3. In the evaluation step of the ROC curves, AUC values of the upregulated circRNAs are in the acceptable range while AUC values of the downregulated circRNAs are in the excellent range. This difference may need to be discussed.

Reviewer 2 ·

Basic reporting

no comment

Experimental design

In order to explain the study design a flow chart should be added. Therefore the outline of the study design would become more clear. Especially the databases used for CircRNA-miRNA-mRNA Interaction Prediction, pathway analysis, Protein-Protein Interaction (PPI) Network Construction, GEPIA should be explained.

Validity of the findings

Power analysis should be performed in order to conclude whether the total sample size is sufficient for comparison.

Additional comments

Limitations of the study which is mentioned in the last paragraph of the Discussion Section should be improved. Professional English should be used. The reasons of the limitations should be explained in detail.

Reviewer 3 ·

Basic reporting

In this manuscript the authors undertook to study a potential diagnostic and predictive indicators of several critical circRNA-miRNA-mRNA regulatory axes in CRC.
The manuscript in its current form has major flaws, making it difficult to judge its suitability for publication.
The introduction part should be reorganized. Following are the points those need to be addressed in introduction :
• Some statements should be double checked for accuracy and added more references.
• Relevant prior literature should be appropriately referenced about relationship between circRNAs and colerectal cancer. Please, cite the manuscripts entitled, " circCAMSAP1, CircHIPK3 and CircRNA_0000392 etc in the introduction.

Figure 1 represents the data quality of each chip is reliable after data normalization. If necessary, it can be given given in the supplementary file section.

Figure 5 indicates that Correlation of six candidate circRNAs with clinicopathological parameters is not statistically significant. Also in the data stardard deviation is too high. My concern is that is it correct to refer this study as six Circular RNAs as diagnostic markers for colorectal cancer. Kindly explain how?

Figure 6 and 8 are graphs showing The basic structural patterns of differantional expressed circRNA and miRNA binding secondary structure. These can be also given in the supplementary file section. I don't think it's necessary in the main figure.

In Figure 7 entitled “The top five targeted miRNAs for circRNA-miRNA regulatory network”, the tumor suppressor or oncogenic function of miRNAs in the CRC should be discussed in the discussion part. Although there is limited information about circRNA, the function of miRNAs are well known. For example, PMID: 34822025 and PMID: 30962766 are two different studies about miR-665 in CRC recently.

Figure 9 could be organized in a better way to make it easier for readers to understand the data.

Figure 13 "cirReAXEs in the CRC" shows that hsa-circRNA-102293 target same miRNA family (miR-30b-5p and miR-30e-5p). Also same trend are also between miRNA-mRNA. How can you explain it, please your comment add to discussion part.

Experimental design

Hypothesis or aim of this study is unclear. Please explain in end of the introduction section.

It is recommended to deposit your circRNA microarray data in Gene Expression Omnibus (GEO) or ArrayExpress and accession numbers could be provided in the published manuscript.

Validity of the findings

The discussion and conclusion parts of the manuscript need to be re-written. There are no clear concluding remarks at the end of discussion. Discussion needs to be recomposed with clear-cut final conclusion that marks the importance of all the analyses done by the authors.

Kindly, explain the future perspective of the study in more elaborate manner, if possible.

---

## Round 0.2 · Minor Revisions

Please address all issues raised by the reviewers.

·

Basic reporting

The authors were used clear English throughout the manuscript, however still there are some spelling mistakes. These minor mistakes should be corrected in the manuscript. The main structure of the article, figures, tables and references are acceptable.

Experimental design

The authors have tried to complete all the suggested revisions in the manuscript. However, as an answer of the suggestion “In the results and discussion parts circRNAs differential expression patterns should be considered between colon and rectum cancer cases.” they have only made a general supplementary discussion about the difference between colon cancer and rectum cancer. It will be better if they made this dicsussion by using their own data.

Validity of the findings

It is acceptable

Reviewer 3 ·

Basic reporting

Research question is not well defined. Please add your hypothesis at the end of the introduction part, in which it is stated how your research fills an identified knowledge gap.

Figure 5 indicates that Correlation of six candidate circRNAs with clinicopathological parameters is not statistically significant. Also in the data stardard deviation is too high. My concern is that is it correct to refer this study as six Circular RNAs as diagnostic markers for colorectal cancer. Your answer supports this concern. Just showing the same expression trend, microarray and qRT-PCR analysis, is not sufficient to be identified as a diagnostic marker. Therefore, I think the figure should be removed in the manuscript.

Experimental design

no comment

Validity of the findings

The Discussion and Conclusion sections of the manuscript need to be re-written. Following are the points those need to be addressed :

The sentences between lines 352-366 are related to the general information and belong to the introduction part. This sentences which affected the flow of this part should be removed. Also, I think there should be better transitions into the next topics.

Sentences starting with “It should be pointed out that ----” in line 448 and to the end of the paragraph, these sentences are not suitable for a scientific manuscript. Please re-written.

The conclusion part should be re-written. Sentences starting with “CircRNA has important-----” in line 478 belong to the conclusion part. I recommend that you end of this paragraph with a sentence that connects with your own data.

---

## Round 0.3 · accepted · Accept

All of the issues raised by the reviewers are now answered.